# Severe Acquired Brain Injury: Prognostic Factors of Discharge Outcome in Older Adults

**DOI:** 10.3390/brainsci12091232

**Published:** 2022-09-12

**Authors:** Augusto Fusco, Caterina Galluccio, Letizia Castelli, Costanza Pazzaglia, Roberta Pastorino, Denise Pires Marafon, Roberto Bernabei, Silvia Giovannini, Luca Padua

**Affiliations:** 1UOC Neuroriabilitazione ad Alta Intensità, Fondazione Policlinico Universitario A. Gemelli IRCCS, 00168 Rome, Italy; 2Department of Geriatrics and Orthopaedics, Università Cattolica del Sacro Cuore, 00168 Rome, Italy; 3Department of Aging, Neurological, Orthopaedic and Head-Neck Sciences, Fondazione Policlinico Universitario A. Gemelli IRCCS, 00168 Rome, Italy; 4Department of Woman and Child Health and Public Health—Public Health Area, Fondazione Policlinico Universitario A. Gemelli IRCCS, 00168 Rome, Italy; 5Section of Hygiene, University Department of Life Sciences and Public Health, Università Cattolica del Sacro Cuore, 00168 Rome, Italy; 6UOS Neuroriabilitazione Post-acuzie, Fondazione Policlinico Universitario A. Gemelli IRCCS, 00168 Rome, Italy

**Keywords:** acquired brain injury, cognitive function, rehabilitation, personalized medicine, prognostic factors, setting, discharge, elderly, device, disorder of consciousness

## Abstract

Severe Acquired Brain Injury (sABI) is a leading cause of disability and requires intensive rehabilitation treatment. Discharge from the rehabilitation ward is a key moment in patient management. Delays in patient discharge can adversely affect hospital productivity and increase healthcare costs. The discharge should be structured from the hospital admission toward the most appropriate environment. The purpose of our study is to investigate early predictors of outcome for discharge in older adults with sABI. A retrospective study was performed on 22 patients who were admitted to an intensive neurorehabilitation unit between June 2019 and December 2021. Patients were divided into two outcome categories, good outcome (GO) or poor outcome (PO), based on discharge destination, and the possible prognostic factors were analyzed at one and two months after admission. Among the factors analyzed, changes in the Disability Rating Scale (DRS) and Level of Cognitive Functioning (LCF) at the first and second month of hospitalization were predictive of GO at discharge (DRS, *p* = 0.025; LCF, *p* = 0.011). The presence of percutaneous endoscopic gastrostomy at two months after admission was also significantly associated with PO (*p* = 0.038). High Body Mass Index (BMI) and the presence of sepsis at one month after admission were possible predictors of PO (BMI *p* = 0.048; sepsis *p* = 0.014). An analysis of dynamic predictors could be useful to guarantee an early evaluation of hospital discharge in frail patients with sABI.

## 1. Introduction

Severe acquired brain injury (sABI) encompasses a wide range of neurological diseases with a significant impact in terms of mortality and morbidity [1]. Worldwide, it causes over 12 million deaths every year and is the leading cause of disability [2,3]. The rehabilitation of patients with sABI involves a variety of interventions, delivered in a customized manner by a multidisciplinary team [4,5]. 

The clinical outcome can vary, ranging from mild reversible symptoms to lasting impairments, significantly influencing the ability to live independently and the quality of life [6]. The management of these patients should consider achievable rehabilitation outcomes [7]. Their prognosis is dependent not only on the severity of the neurological condition, but also on the presence of comorbidities and medical complications, both influencing survival and disability [8,9]. 

Several studies have been conducted over the years to identify clinical, neurophysiological, radiological, and intrinsic early predictors. Studies on clinical and functional status at discharge from rehabilitation units have focused on single-point assessments. A wide scientific consensus identifies younger age, shorter time to admission into a rehabilitation program, and traumatic etiology as possible factors associated with greater functional recovery [8,10,11]. In older adults, clinical and functional characteristics, such as cognitive impairment, multimorbidity, poor nutritional status, pre-trauma reduced mobility, sarcopenia, and polypharmacy, may hinder the likelihood of achieving positive outcomes during rehabilitation [12,13,14].

In this study, we evaluated early predictors of discharge outcome among older adults with sABI based on their clinical and functional status during the first phase of their admission to a neurorehabilitation unit. The dynamic variations of the parameters analyzed over a period of time, as well as the rapidity with which these variations occur, may enable the early identification of the potential evolution of these patients. This could help address rehabilitation treatment, formulate prognoses more linked to functional status, educate patients’ families and caregivers, and communicate with all healthcare professionals regarding possible discharge settings, thus improving the management of hospital resources.

## 2. Materials and Methods

### 2.1. Study Design

This preliminary study is part of a prospective, longitudinal, observational cohort study conducted in an intensive neurorehabilitation unit (INRU), currently in progress. This study was conducted in accordance with the International Guidelines for Good Clinical Practice and the Declaration of Helsinki. Prior to participating, all subjects or their legal caregivers provided written informed consent. 

From June 2019 to December 2021, we retrospectively assessed patients with sABI admitted to INRU. The following criteria were required for inclusion: diagnosis of sABI with a Glasgow Coma Score (GCS) of less than 8, lasting more than 24 h after the injury, and requiring intensive neurorehabilitation care; age greater than 70 years. The following patients were excluded from participation: patients who did not have informed consent or those who did not complete at least one month of hospitalization—due to sudden deaths or to transfers in other wards due to clinical complications—because that made it impossible to carry out a dynamic prognostic assessment based on changes over time.

Due to the preliminary report, we did not perform a sample size estimation. The clinical assessment and the filling of the rating scales were carried out by a multidisciplinary group of professionals (physicians, neuropsychologists, physiotherapists, speech therapists, occupational therapists, and nurses). A weekly meeting was scheduled as part of the clinical practice in order to guarantee homogeneity in the clinical and therapeutic course.

### 2.2. Timing of Assessment

A four-step assessment was performed: at the time of admission into the INRU (baseline, T0), at the end of the first month (T1), at the end of the second month (T2), and at discharge only as a record of discharge setting. The timeline of the study is presented in Figure 1.

### 2.3. Discharge Outcome

We clinically assessed inpatients of the INRU and divided them into two groups of outcomes: good (GO) or poor (PO). 

Patients with successful rehabilitation treatment (GO) were those who, after discharge from INRU, were transferred to a rehabilitation ward at a lower intensity or to home. On the other hand, we defined patients with unsuccessful treatment (PO) as those who were discharged to a residential health facility (long-term care facility) or to a hospice (palliative care), or who died during hospitalization (after two months from admission). 

We excluded from the statistical analysis the cases in which the discharge setting did not match with the clinical outcome at discharge as assessed in the study design. In particular, patients who were transferred to a residential healthcare facility representative of a PO were excluded when the transfer was due not to clinical reasons but to logistical/family/management issues. For the same reason, patients discharged home, the setting of a GO, were not included in the analysis if transfer was due to family/private willingness despite their poor clinical conditions.

### 2.4. Assessment

At T0, demographical and clinical data were collected. In particular, the following variables were detected: gender, age, general clinical conditions, time from the injury, sABI etiology, lowest GCS recorded in acute care division, body mass index (BMI), and presence of disorder of consciousness (DOC), included as unresponsive wakefulness syndrome (UWS) and the minimally conscious state (MCS) [15,16]. Furthermore, the possible occurrences of complications, such as sepsis, pressure ulcers, and critical illness myopathy and neuropathy (CRIMYNE) syndrome, were recorded. Patients were considered to have coexisting comorbidities if any of the following diseases were present: hypertension, diabetes, cardiac disorder, psychiatric disorder, renal failure, chronic obstructive pulmonary disease, or other significant medical problems.

For each patient, we assessed the cognitive and behavioral status by means of the following outcome measures: Level of Cognitive Functioning (LCF) [17,18], the level of global disability assessed through the Disability Rating Scale (DRS) [19,20,21], and the level of independence in the daily life activities as per the modified Barthel Index (mBI) [22,23].

To highlight the patient’s improvement at the different time points of analysis, we used the percentage change in clinical rating scales (DRS, LCF, mBI) as a rehabilitation gradient: between T0 and T1 = (T1 − T0)/T0 × 100; between T0 and T2 = (T2 − T0)/T0 × 100; and between T1 and T2 = (T2 − T1)/T1 × 100.

The presence of a tracheostomy, nasogastric tube (NGT), percutaneous endoscopic gastrostomy (PEG), central venous catheter (CVC) or peripherally inserted central venous catheter (PICC), and urinary catheter (UC) was registered at all the assessment time-points. Finally, we registered the discharge setting according to the main outcome measure.

### 2.5. Statistical Analysis

Descriptive statistics were conducted to describe the study participants. Qualitative variables were expressed as absolute frequencies and percentages; quantitative variables were reported as medians and ranges. As our study has a small sample size, the evaluation of percentage change after 1 and 2 months of hospitalization is exploratory. However, the percentage change gives us an indication of the trend of the scales of rehabilitation, facilitating the identification of patients most at risk of a poor outcome during follow-up. The measure “percentage change” has already been adopted in other studies with rating scales [24,25,26].

All statistical tests were two-sided; a *p*-value < 0.05 was considered statistically significant. Statistical analysis was performed using Stata software (StataCorp. 2017. Stata Statistical Software: Release 15.1., StataCorp LP, College Station, TX, USA).

## 3. Results

During the study period, 104 patients were admitted to INRU (Figure 2).

A total of 27 patients (59.3% males) with a median age of 77.3 years (range 70.1–86.2) were enrolled according to the inclusion/exclusion criteria; 22 patients were included in the statistical analysis at T1 and 19 in the analysis at T2. 

The etiologies of the sABI were: traumatic (*n* = 3), vascular (*n* = 12; 6 ischemia, 6 hemorrhagia), anoxic (*n* = 5), and miscellaneous (*n* = 7, due to cerebral infections including SARS-CoV2 and local tumor). The median time from the acute event was 39 days (range 8–404). The median of the BMI was 25.0 kg/m^2^. Ten patients presented DOC at T0, and in five patients, the CRIMYNE was also concomitant.

Demographic and clinical characteristics of the sample size at baseline are reported in Table 1.

### 3.1. Predictors of Good/Poor Outcomes at 1 Month

Twenty-two inpatients had a 1-month hospitalization period in the INRU (15 presented GO and 7 PO). No statistically significant differences between the two groups were found for sex, age, the time elapsed between the acute event and admission, and the total length of stay. A mildly statistically significant higher BMI was found in the patients with a PO (*p* = 0.048). 

A statistically significant higher probability of a GO was achieved in patients who presented from T0 to T1 an improvement in terms of LCF and DRS (*p* = 0.011 and *p* = 0.025, respectively). The mBI changes were close to the significance threshold (*p* = 0.057). Sepsis was significantly associated with a higher probability of a PO (five patients versus two patients with a GO; *p* = 0.014). The presence of pressure ulcers was close to the significance threshold (*p* = 0.074), even if the number of patients with poor and good outcomes was the same (*n* = 6).

The presence of DOC at T0 was negatively associated with the outcome (even if only close to the significance, *p* = 0.052), while the presence of CRIMYNE was not significant in influencing the outcome.

No other studied factors were statistically associated (tracheostomy, NGT or PEG, CVC or PICC, UC, and the presence of other comorbidities).

The results are reported in Table 2. 

### 3.2. Predictors of Good/Poor Outcomes at 2 Months

Nineteen patients had a 2-month hospitalization period in the INRU (fourteen patients achieved a GO and five a PO).

No significant differences were found between the two groups regarding the time elapsed between the acute event and admission and the total length of stay. Moreover, the presence of pressure ulcers and CRIMYNE syndrome did not statistically influence the outcome at discharge. In addition, the number of comorbidities was not statistically related.

With respect to predictors at 1 month, both BMI and the presence of sepsis were not significantly different (*p* = 0.12 and *p* = 0.11, respectively).

At T2, seven patients who presented DOC at T0 were significantly influenced toward a PO at discharge (80% of patients of the group) (*p* = 0.038). 

An improvement in cognitive status and global disability was still significantly associated with a GO (LCF: *p* = 0.049; DRS: *p* = 0.024). For both outcomes, the improvement was mainly related to the outcome achieved in the first month, being not significantly different between T1 and T2 (Table 3).

No statistically significant difference in the mBI at discharge was found between the two groups (*p* = 0.16), despite the fact that we found a significant improvement between the first and second month (*p* = 0.038). 

Finally, a statistically significant difference between the groups was found for the use of PEG. PEG was used for 80.0% of patients with poor outcomes (4 out of 5 patients) compared with 21.4% of patients with GO (3 out of 14 patients) (*p* = 0.038).

## 4. Discussion

In older adults with sABI, the choice of discharge setting is critical in rehabilitation management. The measurements that can aid in predicting a patient’s destination after discharge may facilitate the complexity of management of those patients by preparing families and caregivers psychologically and logistically [27,28,29].

The novelty of our approach was to find out early predictors of discharge outcome taking into consideration not only a one-time-point assessment but also the entire trajectory of the patient’s hospitalization after the first and second months from admission. Our results reveal that the improvement observed in the first month of admission (as shown by the results in the gradient of DRS and LCF scales between admission and T1) was associated with a higher probability of a GO. On the contrary, the early failure to improve in terms of global disability (as evidenced by mBI) negatively impacts the achievement of a good discharge outcome. Statistically, we observed a strict tendency to have no progress in the first and second months of hospitalization in cases of patients with poor outcome at discharge.

Globally, an advanced age is considered a negative prognostic factor for the functional recovery of patients with sABI [30,31,32]. A recent cohort study on patients with a traumatic brain injury over a period of thirty years showed that functional recovery of the disorder of consciousness is better in younger individuals after an intensive rehabilitation treatment [13]. 

The presence of DOC at admission negatively influenced early improvement and discharge outcome in our sample. Surprisingly, the presence of devices (tracheostomy, NGT, CVC or PICC, and UC), the occurrence of CRIMYNE, and the number of morbidities did not significantly affect discharge outcome. 

Conversely, the presence of PEG was significantly associated with a negative outcome. PEG is considered an excellent device for the long-term management of patients with dysphagia, unable to feed themselves, with an optimal patient comfort, less frequent complications, and greater improvements in nutritional status [33,34]. PEG could be a unique choice in the maintenance of good nutritional status in safety conditions [35]. At the same time, it is also possible that PEG could negatively influence the quality of life of patients due to the underlying chronic illness [36]. We can assume that is not PEG that negatively affects the outcome but the severity of the condition that forces the use of PEG. 

Finally, a high BMI and the presence of sepsis in the first month significantly reduced the possibility of achieving a favorable outcome. These conditions were not significantly different at T2. We believe that this could be due to the small sample size. They could represent early predictors of PO in the future, but we need more data to confirm this consideration. 

The possible limitations of the study should be taken into account when considering our results. The small sample size did not allow for analyzing the influence of the etiology on the outcome. Nevertheless, because of the age evaluated, it is more probable to have patients with non-traumatic sABI. Kowalski and colleagues have declared a potential referral bias in studies examining inpatient rehabilitation for patients with TBI, who are younger and have a potentially better prognosis [13]. Moreover, the small sample size highlights an important variability between the time of the acute event and the admission into the INRU, depending on the patient’s clinical condition. In accordance with our national health system’s criteria to be admitted to this type of ward, the patient has to be clinically stable, and this condition can require a long time.

Another limit is the lack of diversification of DOC in UWS and MCS due to the small sample. A possible 40% misdiagnosis rate has already been reported in specialized centers due to the difficult distinction between UWS and MCS [37].

Lastly, we did not include a long-term follow-up assessment. Our last evaluation was scheduled at discharge, but only as a discharge setting record. Longitudinal studies suggest continued functional improvement for several years after rehabilitation discharge [38,39]. The aim of our study was to find out prognostic factors of discharge and not regarding functional recovery [40]. Finally, we did not evaluate neuroanatomic injury through second-level imaging investigations, such as CT and MRI, or other prognostic factors, such as neurophysiology and laboratory tests, in accordance with the aim of our study. 

Recently, new methods for diagnosis and prognosis, based on progresses in machine learning for prognostic factors, biomarker evaluation, and disease prediction, have been proposed [41,42,43,44]. Machine learning can automatically acquire and analyze real-time data and develop models that assist clinicians in making decisions in their clinical practice [45]. However, these models have not yet been widely implemented in clinical practice [46]. Currently, several algorithms have been tested. These models are based on small data sets (<200 events) to discriminate between patients with good and poor outcomes [47], but it remains unclear what the impact on prognostication is due to difficulties in calibration and discrimination [46,48]. Furthermore, the method and results of our study along with the use of advanced neurophysiologic techniques such as quantitative electroencephalography may contribute to more accurate diagnoses and prognoses [49,50].

All of these limitations represent the basis for other, larger, multicenter studies. We also note that the rapid increase in the large number of studies needs to be confirmed with systematic reviews and meta-analyses to provide stronger scientific evidence. 

## 5. Conclusions

The prediction of the possible evolution of patient pathways and outcomes may lead to an easier and earlier discharge once the intensive rehabilitation process is over. In elderly people, the outcome achieved at discharge is related to the changes in functional and consciousness state obtained in the first month of rehabilitation. PEG was used in a higher percentage of patients who achieved a poor outcome. The presence of sepsis and a high BMI in the first month of hospitalization could be also predictors of a poor outcome, but more data and a larger sample size are necessary to confirm this result. 

In the future, it would be interesting to evaluate whether these early dynamic predictors of outcome at discharge could be also associated with a patient’s functional outcome in terms of residual disability and functional capacity.

## Figures and Tables

**Figure 1 brainsci-12-01232-f001:**
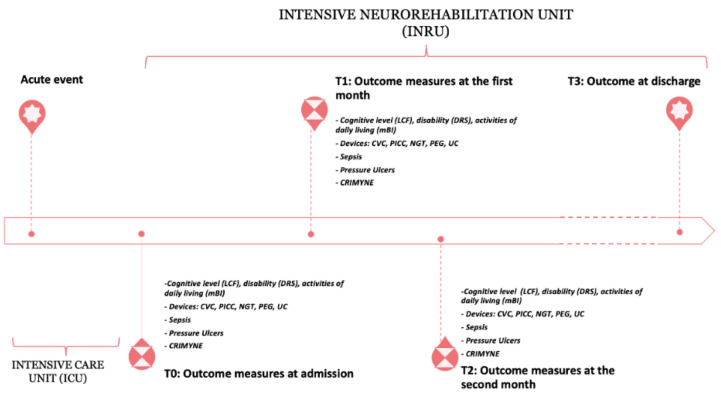
Detailed experimental protocol and timeline of the study. LCF = Levels of Cognitive Functioning Scale; eight ratings levels (from no response to appropriate response), with a good test–retest and inter-rater reliability and concurrent and predictive validity. DRS = Disability rating scale; a 30-point scale and 8-item measure providing information on the level of disability of patients with sABI. mBI = modified Barthel Index; a 10-item instrument measuring functional independence in personal activities of daily living. Presence of devices (yes/no): CVC = central venous catheter; PICC = peripherally inserted central catheter; NGT = nasogastric tube; PEG = percutaneous endoscopic gastrostomy; UC = urinary catheter. Sepsis (yes/no). CRYMINE = critical illness myopathy and/or neuropathy (yes/no).

**Figure 2 brainsci-12-01232-f002:**
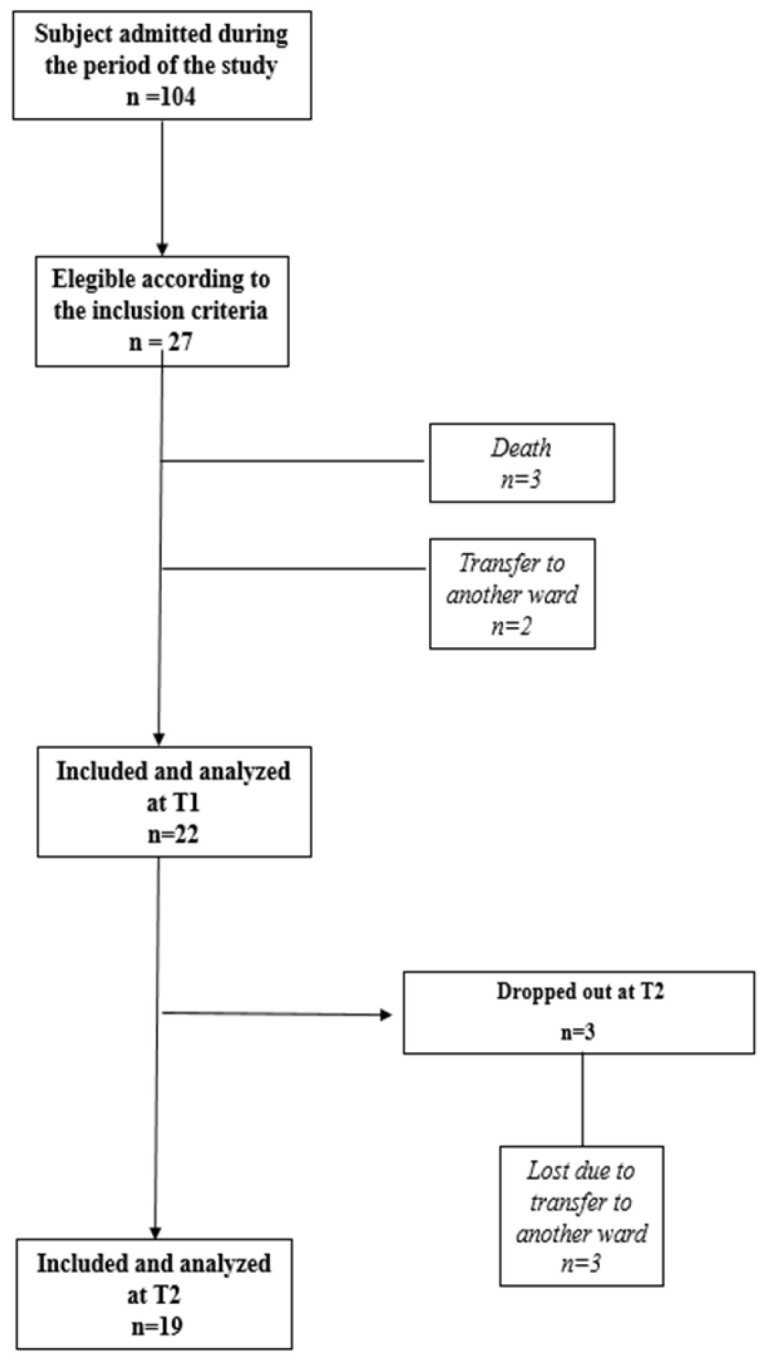
Flow-chart diagram of the study.

**Table 1 brainsci-12-01232-t001:** Patient features at baseline (N = 27).

Age at Injury, Median (Range), y	77.3 (Range 70.1–86.2)
Cause of sABI, *n* (%)	
Traumatic	3 (11.1)
Stroke (ischemic/hemorrhagic)	12 (44.4)
Hypoxic (heart arrest)	5 (18.5)
Infections/Tumor	5 (18.5)
Post COVID-19	2 (7.4)
Elapsed time from event to INRU (days), median (range)	39 (8–404)
Length of stay (days), median (range)	82 (11–357)
DOC, *n* (%)	10 (37.0)
CRIMYNE, *n* (%)	5 (18.5)
Sepsis, *n* (%)	15 (55.6)
Pressure ulcers, *n* (%)	15 (55.6)
Tracheostomy, *n* (%)	19 (70.4)
Days, median (range)	50 (10–205)
NGT, *n* (%)	15 (55.6)
Days, median (range)	43 (7–113)
PEG, *n* (%)	9 (33.3)
Days, median (range)	114 (12–235)
CVC/PICC, *n* (%)	15 (55.6)
Days, median (range)	37 (2–89)
UC, *n* (%)	26 (96.3)
Days, median (range)	72 (11–357)
Comorbidity, *n* (%)	26 (96.3)
Numbers of comorbidity, median (range)	3 (1–5)

INRU: intensive neurorehabilitation unit; DOC: presence of disturbance of consciousness (on admission to the ward); CRIMYNE: critical illness polyneuropathy and/or myopathy; NGT: nasogastric tube; PEG percutaneous endoscopic gastrostomy; CVC/PICC: central venous catheter/peripherally inserted central venous catheter; UC: urinary catheter.

**Table 2 brainsci-12-01232-t002:** Results of the analyzed variables at T1 assessment (N = 22). In bold are the significant factors and their *p*-values. In italics are the factors and their *p*-values that resulted in close-to-threshold levels of significance at statistical analysis (*p* < 0.05).

	All N = 22	Good Outcome N = 15	Poor Outcome N = 7	*p*-Value *
Male, *n* (%)	12 (54.6)	9 (60.0)	3 (42.9)	0.65
Age (years), median (range)	77.5 (70.1–86.2)	77.3 (72.4–85.6)	78.5 (70.1–86.2)	1.0
Elapsed time from event to ICRU (days), median (range)	39 (8–153)	39 (8–93)	69 (25–153)	0.22
Length of stay (days), median (range)	98 (19–357)	93 (41–357)	103 (19–205)	0.99
**Body Mass Index (BMI), median (range)**	25.2 (20.1–29.3)	24.8 (20.1–28.6)	25.9 (24.2–29.3)	**0.048**
Cause of injury, *n* (%)				-
Trauma	3 (13.6)	2 (13.3)	1 (14.3)	-
Stroke	9 (40.9)	4 (26.7)	5 (71.4)	-
Hypoxic (heart arrest)	4 (18.2)	4 (26.7)	0 (0.0)	-
Infection/Tumor	4 (18.2)	3 (20.0)	1 (14.3)	-
Post COVID-19	2 (9.1)	2 (13.3)	0 (0.0)	-
*DOC*, *n* (%)	8 (36.4)	3 (20.0)	5 (71.4)	*0.052*
CRIMYNE, *n* (%)	5 (22.7)	4 (26.7)	1 (14.3)	1.0
Rehabilitation Gradient: (T1 − T0/T0) × 100				
**DRS, median (range)**	−3.9 (−75.0–38.1)	−5.9 (−75.0–0.0)	0.0 (−5.3–38.1)	**0.025**
**LCF, median (range)**	0.0 (−33.3–166.7)	33.3 (−33.3–166.7)	0.0 (−33.3–0.0)	**0.011**
mBI, median (range)	0.0 (0.0–550.0)	0.0 (0.0–550.0)	0.0 (0.0–0.0)	*0.057*
**Sepsis T0-T1, *n* (%)**	7 (31.8)	2 (13.3)	5 (71.4)	**0.014**
*Pressure Ulcers*^#^, *n* (%)	12 (54.6)	6 (40.0)	6 (85.7)	*0.074*
Devices (T0-T1), *n* (%)				
Tracheostomy	10 (45.5)	6 (40.0)	4 (57.1)	0.65
NGT	9 (40.9)	7 (46.7)	2 (28.6)	0.65
PEG	7 (31.8)	3 (20.0)	4 (57.1)	0.15
CVC/PICC	7 (31.8)	3 (20.0)	4 (57.1)	0.15
UC	18 (81.8)	13 (86.7)	5 (71.4)	0.57
Comorbidity, *n* (%)	21 (95.5)	15 (100)	6 (85.7)	0.32
Numbers of Comorbidity, median (range)	2 (1–5)	2 (1–4)	3 (1–5)	0.81

* Fisher’s exact test; Mann–Whitney test. ^#^ At any time during admission. Bold: The factors resulted significant for the study.

**Table 3 brainsci-12-01232-t003:** Results of the analyzed variables at T2 assessment (N = 19). In bold are the significant factors and their *p*-values.

	All N = 19	Good Outcome N = 14	Poor Outcome N = 5	*p*-Value *
Male, *n* (%)	11 (57.9)	8 (57.1)	3 (60.0)	1.0
Age (years), median (range)	76.0 (70.1–85.6)	76.7 (72.4–85.6)	73.9 (70.1–80.7)	0.26
Elapsed time from event to ICRU (days), median (range)	39 (8–153)	37 (8–93)	69 (27–153)	0.16
Length of stay (days), median (range)	105 (50–357)	99 (52–357)	114 (50–205)	0.67
Body Mass Index (BMI), median (range)	25.0 (20.1–29.3)	24.9 (20.1–28.6)	25.9 (24.2–29.3)	0.12
Cause of injury, *n* (%)				-
Trauma	2 (10.5)	1 (7.1)	1 (20.0)	-
Stroke	8 (42.2)	4 (28.5)	4 (80.0)	-
Hypoxic (heart arrest)	4 (21.1)	4 (28.6)	0 (0.0)	-
Infection/tumor	3 (15.8)	3 (21.4)	0 (0.0)	-
Post COVID-19	2 (10.5)	2 (14.3)	0 (0.0)	-
**DOC, *n* (%)**	7 (36.8)	3 (21.4)	4 (80.0)	**0.038**
CRIMYNE, *n* (%)	5 (26.3)	4 (28.6)	1 (20.0)	1.0
Rehabilitation gradient: (T2 − T0/T0) × 100				
**DRS, median (range**)	−16.7 (−58.3–14.3)	−17.5 (−58.3–−4.8)	0.0 (−17.2–14.3)	**0.024**
**LCF, median (range)**	50.0 (−16.7–166.7)	58.3 (−16.7–166.7)	0.0 (0.0–50.0)	**0.049**
mBI, median (range)	0.0 (−33.3–1000.0)	10.0 (−33.3–1000.0)	0.0 (0.0–0.0)	0.16
Rehabilitation gradient: (T2 − T1/T1) × 100				
DRS, median (range)	−12.5 (−58.3–0.0)	−12.5 (−58.3–0.0)	−11.5 (−17.2–0.0)	0.42
LCF, median (range)	0.0 (0.0–100.0)	0.0 (0.0–100.0)	0.0 (0.0–50.0)	1.0
mBI, median (range)	20.0 (−50.0–1100.0)	55.0 (−50.0–1100.0)	0.0 (0.0–0.0)	**0.031**
Sepsis (T0-T2), *n* (%)	8 (42.1)	4 (28.6)	4 (80.0)	0.11
Pressure Ulcers ^#^, *n* (%)	10 (52.6)	6 (42.9)	4 (80.0)	0.30
Devices (T0-T2), *n* (%)				
Tracheostomy	13 (68.4)	8 (57.1)	5 (100)	0.13
NGT	10 (52.6)	8 (57.1)	2 (40.0)	0.63
**PEG**	7 (36.8)	3 (21.4)	4 (80.0)	**0.038**
CVC/PICC	10 (52.6)	6 (42.9)	4 (80.0)	0.30
UC	19 (100)	14 (100)	5 (100)	-
Devices (T1-T2), *n* (%)				
Tracheostomy	8 (42.1)	5 (35.7)	3 (60.0)	0.60
NGT	3 (15.8)	3 (21.4)	0 (0.0)	0.53
**PEG**	7 (36.8)	3 (21.4)	4 (80.0)	**0.038**
CVC/PICC	3 (15.8)	1 (7.1)	2 (40.0)	0.16
UC	14 (73.7)	10 (71.4)	4 (80.0)	1.0
Comorbidity, *n* (%)	18 (94.7)	14 (100)	4 (80.0)	0.26
Numbers of Comorbidity, median (range)	3 (1–5)	3 (1–4)	2 (1–5)	0.99

* Fisher’s exact test; Mann–Whitney test. ^#^ At any time during admission. Bold: The factors resulted significant for the study.

## Data Availability

Not applicable.

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
