# Peer review of "Severe Acquired Brain Injury: Prognostic Factors of Discharge Outcome in Older Adults"

_brainsci, 2022, doi:10.3390/brainsci12091232_

Round 1
Reviewer 1 Report
The retrospective study analyzed dynamic predictors acts as an early evaluation of hospital discharge in frail patients with SABI, demonstrated some correlation with outcome. Here are some suggestions as below
Introduction
The introduction had provided sufficient background knowledge and illustrated the purpose of the study, however, I found the paragraph should be shortened.
Line 49-61 These paragraphs were similar in content, should be simplified.
Lin4 73-91 These paragraphs were also similar in content, should be simplified.
Materials and Methods
Line 97-100
If the study approved by the local institutional review board? Please provide the IRB number.
Line 102-103
This preliminary study seems to be a part of a longitudinal cohort study, and all the data collection should be prospectively retrieved. However, the authors stated that the data was retrospective assessed. I suggested modified the description to illustrate the study is a part of a prospective study.
Line 103
“diagnosis sABI” the grammar is incorrect, please modify
Line 106
Why did the author set an exclusion criteria “at least one month of hospitalization”? Though we expect fulfill inclusion criteria may encounter a prolonged hospitalization course, is there any possibility that a recover within one month? The author should address the reason to set up one month.
Line 111-112
“All team participants were daily involved in the rehabilitative care as part of the neurorehabilitation unit.” Was this information important to the study? The author may consider remove.
Figure 1
Please remove the spell check of “Outcome of discharge”
Line 126 and 128
The category of outcome should be “Defined” instead of “Considered”.
Line 132-134
This paragraph is not clear, and the grammar of the sentence should be modified. And please provide the explanation why would those patients being remove from analysis?
The 2.4 Assessment should be modified since the paragraph is not easy to read. For example in Line 145-149, these paragraphs were related. The author shall rearrange the content to be coherent and fluent.
Line 167-168
The author use percentage change in clinical rating scales (DRS, LCF, mBI) as rehabilitation gradient. However, the scales not totally continuous variable. The author should not manage the improvement by percentage, it is not logical. Please consider using other method to express the improvement.
Table 1
Elapsed time from event to INRU, why would the patient being admitted to INRU at 404 days after the acute event? (the longed period showing in the range)
And in the Table 2 the longest elapsed time from event to ICRU is 153, is there any discordancy?
3.1 Predictors of Good/Poor Outcomes at 1 month and 3.2 Predictors of Good/Poor Outcomes at 2 months
The paragraph should be rearranged for better reading.
Result
Overall, the result are too long. Please shortened
Especially for the paragraph within Line 255-264 and Line 270-293.
Also, the sentence in Line 265-266 did not provide important information, should be simplified and integrated into other paragraph if the author wish to remain.
Conclusion
The conclusion shall be condensed.
Author Response
We would like to thank the Editor and the Reviewers for their time and the valuable suggestions and indications about our study. We have carefully taken into account all the criticisms of the Reviewers to improve our manuscript in this revised version. We have performed an extensive revision of the written language throughout the manuscript. For the sake of clarity, we did not report all the stylistic modification in the text. In the following our point-by-point detailed responses. In Italic, it is reported the changes in the text.
REVIEWER 1
The retrospective study analyzed dynamic predictors acts as an early evaluation of hospital discharge in frail patients with SABI, demonstrated some correlation with outcome. Here are some suggestions as below
Introduction
The introduction had provided sufficient background knowledge and illustrated the purpose of the study, however, I found the paragraph should be shortened.
Line 49-61 These paragraphs were similar in content, should be simplified.
Lin4 73-91 These paragraphs were also similar in content, should be simplified.
AUTHORS: Thank you, these two paragraphs have been extensively modified and condensed as the following.
“Several studies have been conducted over the years to identify clinical, neurophysiological, radiological, or intrinsic early predictors. Studies on clinical and functional status at discharge from rehabilitation units have focused on single-point assessments. A wide scientific consensus identifies younger age, shorter time to admission into a rehabilitation program, and traumatic aetiology as possible factors associated with greater functional recovery [8,10,11]. In older adults, clinical and functional characteristics such as cognitive impairment, multimorbidity, poor nutritional status, pre-trauma reduced mobility, sarcopenia, and polypharmacy may hinder the likelihood to achieve positive outcomes during rehabilitation [12–14].”
Materials and Methods
Line 97-100
If the study approved by the local institutional review board? Please provide the IRB number.
AUTHORS: Thank You, This study was conducted in accordance with the International Guidelines for Good Clinical Practice and the Declaration of Helsinki, all subjects or their legal caregiver gave written informed consent before participation. It was observational and retrospective and no IRB number is required.
Line 102-103
This preliminary study seems to be a part of a longitudinal cohort study, and all the data collection should be prospectively retrieved. However, the authors stated that the data was retrospective assessed. I suggested modified the description to illustrate the study is a part of a prospective study.
AUTHORS: Thank you for the indication. We have better specified in the first part of the paragraph as follows.
“This preliminary study is part of a prospective, longitudinal, observational cohort study conducted in an Intensive NeuroRehabilitation Unit (INRU), currently in progress.”
Line 103
“diagnosis sABI” the grammar is incorrect, please modify
AUTHORS: Thank you for the indication. We have corrected this mistake.
Line 106
Why did the author set an exclusion criteria “at least one month of hospitalization”? Though we expect fulfill inclusion criteria may encounter a prolonged hospitalization course, is there any possibility that a recover within one month? The author should address the reason to set up one month.
AUTHORS: Thank You, we have modified text following the suggestions. We hope now the paragraph is clear:
“…those who did not complete at least one month of hospitalization – due to sudden deaths or to transfers in other wards due to clinical complications – because, it was impossible to carry out a dynamic prognostic assessment, based on changes over time”.
Line 111-112
“All team participants were daily involved in the rehabilitative care as part of the neurorehabilitation unit.” Was this information important to the study? The author may consider remove.
AUTHORS: Thank you for the indication. We have deleted this part as you suggested.
Figure 1
Please remove the spell check of “Outcome of discharge”
AUTHORS: Thank you for the indication. In accordance with your and the other reviewer suggestions, we have revised the figure as you suggested.
Line 126 and 128
The category of outcome should be “Defined” instead of “Considered”.
AUTHORS: Thank you for the suggestion. We have followed your indication and modified the text.
Line 132-134
This paragraph is not clear, and the grammar of the sentence should be modified. And please provide the explanation why would those patients being remove from analysis?
AUTHORS: Thank You, we have modified text following the suggestions. We hope the period is now more readable:
“We excluded from the statistical analysis the cases in which the discharge setting did not match with the clinical outcome at discharge as assessed in the study design. In particular: patients who were transferred to a residential health care facility, that represented a PO, were excluded when the transfer was due not to clinical reasons but to logistical/family/management issues. For the same reason, patients discharged home, setting of a GO, were not included in the analysis if that transfer was not due to clinical reasons but to familiar/private issues”.
The 2.4 Assessment should be modified since the paragraph is not easy to read. For example in Line 145-149, these paragraphs were related. The author shall rearrange the content to be coherent and fluent.
AUTHORS: Thank you for the suggestion. We have modified the text as follows.
“At T0 demographical and clinical data were collected. In particular, the following variables were detected: gender, age, general clinical conditions, time from the injury, sABI etiology, lowest GCS recorded in acute care division; body mass index (BMI); presence of disorder of consciousness (DOC), included as unresponsive wakefulness syndrome (UWS) and the minimally conscious state (MCS) [15,16].”
Line 167-168
The author use percentage change in clinical rating scales (DRS, LCF, mBI) as rehabilitation gradient. However, the scales not totally continuous variable. The author should not manage the improvement by percentage, it is not logical. Please consider using other method to express the improvement.
AUTHORS: All the section related to statistical analysis has been revised in accordance with uor and the other reviewer’s suggestions. In the following, we have reported the modified text:
“Descriptive statistics were conducted to describe the study participants. Qualitative variables were expressed as absolute frequencies and percentages; quantitative variables were reported as medians and range. Since our study has a small sample size, the evaluation of percentage change after 1 and 2 months of hospitalization is exploratory. However, the percentage change gives us an indication of the trend of the scales of rehabilitation, facilitating the identification of patients most at risk of poor outcome during follow-up. The measure “percentage change” has already been adopted in other studies with rating scales [24–26].
Qualitative variables were expressed as absolute frequencies and percentages; quantitative variables were reported as medians and range.
All statistical tests were two sided; a p-value <0.05 was considered statistically significant. Statistical analysis was performed using Stata software (StataCorp. 2017. Stata Statistical Software: Release 15.1. College Station, TX: StataCorp LP).”
Table 1
Elapsed time from event to INRU, why would the patient being admitted to INRU at 404 days after the acute event? (the longed period showing in the range)
And in the Table 2 the longest elapsed time from event to ICRU is 153, is there any discordancy?
AUTHORS: Thank You, for Your consideration. After checking the tables, we can confirm that data are correct and there is no discordancy. In fact, in Table 1 (T0: 27 patients) the value of 404 days is the maximum value of days, referring to a patient who was hospitalized for less than one month (because the patient was died) and cannot be considered at T1 (one month: 22 patients). In Table 2, however, the maximum value of days is different because it refers to another patient.
Moreover, we have added your useful suggestion in Discussion:
“Moreover, the small sample size highlights an important variability between the time of the acute event and the admission in INRU, depending on patient's clinical condition. In accordance with criteria to be admitted to this type of ward, for our National Health System, the patient has to be clinically stable and this condition can require a long time”.
3.1 Predictors of Good/Poor Outcomes at 1 month and 3.2 Predictors of Good/Poor Outcomes at 2 months
The paragraph should be rearranged for better reading.
Result
Overall, the result are too long. Please shortened
Especially for the paragraph within Line 255-264 and Line 270-293.
AUTHORS: Thank you for the comment. All the suggested parts have been extensively modified and condensed as you indicated. For the sake of clarity, we have not reported details (see into the text).
Also, the sentence in Line 265-266 did not provide important information, should be simplified and integrated into other paragraph if the author wish to remain.
AUTHORS: Thank you for the suggestion. In accordance with the comment of the reviewer 2, we have integrated this part in a wider part, as follows:
“Recently, new methods for diagnosis and prognosis have been proposed, based on progresses in machine learning for prognostic factors, biomarker evaluation and disease prediction [41–44]. Machine learning can automatically acquire and analyze real-time data and develop models that assist clinicians in their clinical practice by developing models that help them make clinical decisions [45]. However, these models have not yet been widely implemented in clinical practice [46]. Currently, several algorithms have been tested. These models are based on small data sets (<200 events) to discriminate between patients with good and poor outcomes [47], but it remains unclear what the impact on prognostication is due to difficulties in calibration and discrimination. [46,48]. Further-more, the method and results of our study along with the use of advanced neurophysiologic techniques such as quantitative electroencephalography may contribute to more ac-curate diagnosis and prognosis [49,50].”
Conclusion
The conclusion shall be condensed.
AUTHORS: Thank You, we have modified text following the suggestions.
“The prediction of the possible evolution of the patient's pathway and outcomes may lead to an easier and earlier discharge, once the intensive rehabilitation process is over. In el-derly people, the outcome achieved at discharge is related to the changes of functional and consciousness state obtained in the first month of rehabilitation. PEG was used in a higher percentage of patients who achieved a poor outcome. The presence of sepsis and a high BMI in the first month of hospitalization could be also predictors of poor outcome, but more data and a larger sample size are necessary to confirm this result.
In the future, it would be interesting to evaluate whether these early dynamic predictors of outcome at discharge could be also associated to the patient's functional out-come, in terms of residual disability and functional capacity.”
Reviewer 2 Report
This study aimed to evaluate prognostic factors of discharge outcomes in older adults with severe acquired brain injury. I have the following suggestions.
What is the novelty of this study although several prognostic factors of discharge outcome of severe acquired brain injury patients have been proposed earlier?
Please write down the contribution of the study at the end part of the Introduction section in bulleted form.
Authors should describe studies related to the state-of-art machine-learning based prediction for prognostic factors, biomarker evaluation, and disease prediction, such as, article, healthsos: real-time health monitoring system for stroke prognostics; in article, quantitative evaluation of eeg-biomarkers for prediction of sleep stages; and in article, ECG are investigated in article quantifying physiological biomarkers of a microwave brain stimulation device.
The authors should add a figure of the detailed experimental protocol used in this study.
Detailed statistical methodology of predictors of outcomes needs to be reported.
Authors should describe more details of the factors/variables used in this study.
Authors must make a discussion on the similarities and contradictions of their proposed factors with other recent studies by adding a table in a discussion section.
Author Response
We would like to thank the Editor and the Reviewers for their time and the valuable suggestions and indications about our study. We have carefully taken into account all the criticisms of the Reviewers to improve our manuscript in this revised version. We have performed an extensive revision of the written language throughout the manuscript. For the sake of clarity, we did not report all the stylistic modification in the text. In the following our point-by-point detailed responses. In Italic, it is reported the changes in the text.
REVIEWER 2
This study aimed to evaluate prognostic factors of discharge outcomes in older adults with severe acquired brain injury. I have the following suggestions.
What is the novelty of this study although several prognostic factors of discharge outcome of severe acquired brain injury patients have been proposed earlier?
AUTHORS: Thank you for all the suggestions. We have better specified this part as follows:
“Several studies have been conducted over the years to identify clinical, neurophysiological, radiological, or intrinsic early predictors. Studies on clinical and functional status at discharge from rehabilitation units have focused on single-point assessments. A wide scientific consensus identifies younger age, shorter time to admission into a rehabilitation program, and traumatic aetiology as possible factors associated with greater functional recovery [8,10,11]. In older adults, clinical and functional characteristics such as cognitive impairment, multimorbidity, poor nutritional status, pre-trauma reduced mobility, sarcopenia, and polypharmacy may hinder the likelihood to achieve positive outcomes during rehabilitation [12–14].”
Please write down the contribution of the study at the end part of the Introduction section in bulleted form.
AUTHORS: Thank you for all the suggestions. We have better focused the aim of our study as follows:
“In this study, we evaluated early predictors of discharge outcome among older adults with sABI based on their clinical and functional status during the first phase of their ad-mission to an intensive neurorehabilitation unit. The dynamic variations of the parameters analyzed over a period of time, as well as the rapidity with which these variations occur, may enable the early identification of the potential evolution of these patients. This could help address the rehabilitation treatment, formulate prognoses more linked to the functional status, educate patients' families and caregivers, and communicate with all healthcare professionals regarding possible discharge settings, thus improving the management of hospital resources.”
Authors should describe studies related to the state-of-art machine-learning based prediction for prognostic factors, biomarker evaluation, and disease prediction, such as, article, healthsos: real-time health monitoring system for stroke prognostics; in article, quantitative evaluation of eeg-biomarkers for prediction of sleep stages; and in article, ECG are investigated in article quantifying physiological biomarkers of a microwave brain stimulation device.
AUTHORS: Thank you for the suggestion. In accordance with the comment of the reviewer 1, we have better analyzed the scientific consensus. Consistently, we have added some crucial citations in the reference section. Hence, the text has been modified as follows:
“Recently, new methods for diagnosis and prognosis have been proposed, based on progresses in machine learning for prognostic factors, biomarker evaluation and disease prediction [41–44]. Machine learning can automatically acquire and analyze real-time data and develop models that assist clinicians in their clinical practice by developing models that help them make clinical decisions [45]. However, these models have not yet been widely implemented in clinical practice [46]. Currently, several algorithms have been tested. These models are based on small data sets (<200 events) to discriminate between patients with good and poor outcomes [47], but it remains unclear what the impact on prognostication is due to difficulties in calibration and discrimination. [46,48]. Further-more, the method and results of our study along with the use of advanced neurophysio-logic techniques such as quantitative electroencephalography may contribute to more ac-curate diagnosis and prognosis [49,50].”
The authors should add a figure of the detailed experimental protocol used in this study.
AUTHORS: Thank you for the indication. In accordance with your and the other reviewer suggestions, we have revised the figure 1, including more information regarding the experimental protocol and the used outcome measures.
Detailed statistical methodology of predictors of outcomes needs to be reported.
AUTHORS: All the section related to statistical analysis has been revised in accordance with uor and the other reviewer’s suggestions. In the following, we have reported the modified text:
“Descriptive statistics were conducted to describe the study participants. Qualitative variables were expressed as absolute frequencies and percentages; quantitative variables were reported as medians and range. Since our study has a small sample size, the evaluation of percentage change after 1 and 2 months of hospitalization is exploratory. However, the percentage change gives us an indication of the trend of the scales of rehabilitation, facilitating the identification of patients most at risk of poor outcome during follow-up. The measure “percentage change” has already been adopted in other studies with rating scales [24–26].
Qualitative variables were expressed as absolute frequencies and percentages; quantitative variables were reported as medians and range.
All statistical tests were two sided; a p-value <0.05 was considered statistically significant. Statistical analysis was performed using Stata software (StataCorp. 2017. Stata Statistical Software: Release 15.1. College Station, TX: StataCorp LP).”
Authors should describe more details of the factors/variables used in this study.
AUTHORS: Thank you for the comment. Detailed parts have been added in Methods as you indicated. For the sake of clarity, we have not reported details (see into the text).
Authors must make a discussion on the similarities and contradictions of their proposed factors with other recent studies by adding a table in a discussion section.
AUTHORS: Thank you for the comment. We agree that a stronger scientific evidence is needed. Hence, we have added this sentence in the Discussion:
“All of these limitations represent the basis for other, larger, multicenter studies. We also note that the rapid increase of a wide number of studies needs to be verified with system-atic reviews and meta-analyses to provide stronger scientific evidence.”
Nevertheless, due to the used method and the peculiar sample (elderly people), in addition to the request of reviewer 1 to condense many parts of the text, we are unable to analyze better we did.
Round 2
Reviewer 1 Report
I have no more comment regarding the revision.
Thanks for the revision.
Reviewer 2 Report
Thanks to the authors for clarification of my concerns.